# To Treat or Not to Treat: Public Attitudes on the Therapeutic Use of Antibiotics in the Dairy Industry—A Qualitative Study

**DOI:** 10.3390/ani13182913

**Published:** 2023-09-14

**Authors:** Katharine F. Knowlton, Marina A. G. von Keyserlingk

**Affiliations:** 1School of Animal Sciences, Virginia Polytechnic Institute and State University, Blacksburg, VA 24061, USA; kknowlto@vt.edu; 2Animal Welfare Program, Faculty of Land and Food Systems, The University of British Columbia, 2357 Main Mall, Vancouver, BC V6T 1Z4, Canada

**Keywords:** antimicrobial resistance, therapeutic antibiotic use, dairy cattle, consumer, social license

## Abstract

**Simple Summary:**

A qualitative study was conducted using an online survey platform with participants asked to provide their views on one of three scenarios, that differed in the amount of information, describing a farmer administering a proven antibiotic to a sick cow. While many respondents appeared to be supportive of the farmer’s decision to treat the sick cow, some were concerned when they were presented with information describing the potential link between antibiotic use and the spread of antibiotic resistance. Participants also expressed a strong desire for farmers to accept responsibility for caring for the cow, but, when doing so, to employ practices that reduce the outflow of antibiotics from the farm.

**Abstract:**

This paper describes the views of 779 U.S. residents on questions related to therapeutic antibiotic use in dairy cattle. An online survey was conducted with qualitative (open-ended) questions. Respondents were offered one of three scenarios with varying degrees of information describing a farmer with a sick cow that would benefit from antibiotic therapy. The text replies to the open-ended questions were analyzed by grouping responses with similar comments and identifying patterns or themes. Content analysis showed that many of the participants in this study provided farmers with the social license to treat sick cows with antibiotics; however, some participants commented on the social license not necessarily extending to antibiotic use for growth promotion or prophylactic use. Our findings are not generalizable, but may provide some insight that should be considered when developing policies and practices regarding the use of antibiotics on dairy farms that may promote improved alignment with societal values.

## 1. Introduction

Increased microbial antibiotic resistance is reported globally [1] and primarily attributed to antibiotic use in livestock and humans [2,3]. Antimicrobial use in animal agriculture initially began circa 1940 for the treatment of clinical illness in livestock; however, in the 1950s and 1960s their use broadened to also include growth promotion [4,5]. Microbial resistance to antibiotics was observed shortly after their discovery [6,7]. 

In the U.S., animal agriculture accounts for approximately 70% of all antibiotic usage [8]. There are currently 42 different antibiotics approved for use in animal agriculture in the U.S., with 31 of these considered medically important [8]. This duplication of use has raised concerns given the potential for the transmission of antibiotic-resistant bacteria to humans through direct contact with animals or the ingestion of contaminated food and water [9], a phenomenon that is now well-documented [10,11,12]. Animals treated with antibiotics excrete 70 to 90% of the administered dose intact, and these manures are a potentially important contributor to antibiotic resistance in soil and water environments [13,14,15].

It is the non-judicious use of antibiotics that is the greatest contributor to antimicrobial resistance (reviewed by Zaman et al. [16]). That said, limiting agriculture antibiotic usage has been reported to be the most effective method to reduce antibiotic excretion into the environment [17], with some countries (e.g., The Netherlands, Sweden, Denmark) now legislating reductions in antibiotic usage in agriculture [18], as a means to minimize the risks associated with antibiotic resistance and the associated challenges with human health [19]. Given these concerns, it is not surprising that there is growing public debate regarding antibiotic use in animal agriculture [20].

Clinical disease in farm animal species is usually associated with malaise (see review by Weary et al. [21]) and in some cases has been equated with pain and suffering [22,23] and reduced welfare [24]. Known benefits associated with clinical antibiotic use in farm animals include reducing the number of sick days, resolving infections [25], and improving the odds of recovery [26,27,28]. As an example, clinical mastitis in dairy cattle is a costly disease [29] and a major animal welfare concern for the dairy industry [30,31]. Early diagnosis and timely treatment with antibiotics can improve the welfare of cows and reduce the costs associated with the disease [32]. 

Despite known benefits, some argue that antibiotic usage in livestock systems, regardless of its efficacy, should be severely limited, as only then can antibiotic resistance be curtailed [33]. Recent work, however, provides some evidence that although U.S. citizens reject antibiotic use for the purposes of increasing production, they do expect sick farm animals to be treated (dairy, [34]; pigs, [35]). However, to our knowledge little research has focused on how citizens not associated with agriculture trade-off regarding decisions that may have multiple impacts (e.g., the environment, human and animal welfare). Thus, our aim was to firstly document the views of U.S. participants not affiliated with the dairy industry on antibiotic usage when told that a sick animal could benefit from receiving antibiotics but at the cost of risks associated with the environment, antibiotic resistance, and human health. Secondly, we were interested whether views differed when subjects were provided varying amounts and types of information that required them to trade-off the benefits of providing the antibiotics to the cow and the potential downstream effects of the antibiotics, once excreted, on the environment, antibiotic resistance, and human health.

## 2. Materials and Methods

### 2.1. Recruitment and Consent

Consent forms and recruitment documents outlined methods for maintaining confidentiality and a general description of the study methods and objectives. After consenting to participate, each participant was given additional demographic questions (e.g., income, education, political affiliation). Participants were then invited to respond to two open-ended questions and several multiple-choice and demographic questions via an online platform (Fluid Surveys, www.fluidsurveys.com).

Participation was open to residents of the United States over 18 years of age. Participants were invited online via Mechanical Turk (MTurk, www.mturk.com), a method of data collection that is relatively quick, easy, and yields a diverse [36] and attentive [37] pool of subjects, compared with more traditional subject pools, and has been reported to provide high-quality and reliable data [38]. Participants accessed the online survey using the AmazonPrime site, where they could view the short running titles of a large number of surveys. Those that chose to complete our survey did so by clicking on the short title “The case of therapeutic use of antibiotics in livestock”. Payment for participating was USD0.50, which they received once the survey was completed. 

### 2.2. Survey Design

To characterize respondents, demographic information was collected including age, gender, highest level of education completed, political affiliation, and residence (see supplementary material at https://borealisdata.ca/dataset.xhtml?persistentId=doi:10.5683/SP3/1HCPGA). We received 794 completed surveys. To eliminate multiple responses from the same person, multiple submissions from the same IP address were eliminated.

Participants who agreed to participate were randomly assigned to one of three vignettes that differed in length, amount of information provided, and information delivery mode. Some participants presented with the longer text explanation were also provided access to a short animation video that presented the connection between treating the cow, the excretion of the antibiotics and the induction of antibiotic resistance, and ultimately the link with human health. In the short text vignette, they were told to: *Imagine that a farmer has an animal that he/she is raising for food and that the animal gets sick. The farmer knows that administering antibiotics known to treat this illness could improve the animal’s condition.* In the long text vignette, they were told to: *Imagine that a farmer has an animal that he/she is raising for food and that the animal gets sick. The farmer knows that administering antibiotics known to be effective in treating this illness could improve the animal’s health and welfare but is also aware that some of this antibiotic will enter the environment where it could negatively impact the ecosystem. The farmer also knows that antibiotic resistance in human populations is increasing due to widespread use of antibiotics in the livestock industry.* In the final treatment, respondents were provided the same text as the long vignette, but also had the opportunity to access an animated video that showed the antibiotic entering the cow but also being excreted and flowing into the groundwater.

Initially, we were interested in assessing the support of the respondents for treatment and the effect of the amount and type of information provided; thus, all participants were asked to reply to an open-ended question using 300 words or less: “What do you think the farmer should do?” Finally, two closed-ended question were asked: the participant’s preferences when consuming dairy products (“organic”; “conventional”, defined as non-organic; or “don’t consume dairy products”) and their level of concern for the use of antibiotics in animal farming (“Not concerned at all”, “Somewhat concerned”, or “Very concerned”). However, only 61% responded directly to the question of whether the farmer should or should not treat the cow with antibiotics by answering with a definitive yes/no; we have therefore elected to only summarize the qualitative findings.

### 2.3. Qualitative Analysis

We used the NVivo Qualitative Data Management Program (QSY International Pty Ltd. Version 10, 2012) to analyze the open-ended responses using the methodology proposed by [39]: data reduction was achieved using inductive thematic analyses (coding of information around the common ideas conveyed), data display (information organized to permit conclusions), and conclusion drawing and verification (identification of patterns and themes including the use of confirmatory tactics). This type of analysis is based on the premise that the main themes arise from the responses rather than being determined a priori [34]. 

The primary confirmatory tactic employed was the use of three evaluators, blind to demographic information and treatment, who each independently reviewed the same 50 randomly selected responses to the open-ended questions line-by-line. Each evaluator broke the responses down into smaller segments expressing distinct ideas. These ideas were identified as preliminary codes and the three readers compared results and reconciled any discrepancies. The evaluators then read all responses provided by all participants and coded them accordingly. Often, the participant embedded more than one distinct idea in their response. If so, the same sentence was assigned to multiple codes. 

Upon complete analysis by the three evaluators, inter-evaluator reliability was assessed for each open-ended question, after which the evaluators discussed and resolved any discrepancies in coding. As an example, ‘consumer action’ was initially identified as a code, defined as any reference to empowering consumers to be able to choose products from animals given (or not given) antibiotics. Inter-evaluator reliability for this code was low because one of the evaluators coded relevant responses under a separate code, ‘labelling’. Once ambiguous codes were resolved (in this example, by clarifying the distinction between consumer empowerment and explicit reference to labelling), the first author undertook the final coding. Working together, the authors then aggregated coded responses to themes and selected representative quotes from responses. Quotes representing emerging themes were noted and are provided in the results to help validate authors’ interpretation of the data.

## 3. Results

Our convenience sample of respondents represented 45 states (no respondents reported being from Alaska, North Dakota, South Dakota, Vermont, or Wyoming). The final sample size was 779 respondents, of which 82% self-reported as being less than 44 years old, of which half were 25 to 34 years of age. The majority (~75%) of our participants reported having either an associate’s, bachelor’s, or post graduate degree; consuming dairy products (~90%); and living in a suburban setting (~77%) (See also Table 1).

Four overarching themes emerged from the participants’ responses: Expectations of the farm and farmer; Consumer-/Citizen-focused concerns; Environmental concerns; and Trade-offs. The first three themes were discussed by participants from three of the vignettes, with the last theme only present in the responses from participants that received the long vignette that made reference to the fact that antibiotics can enter the environment via the manure coming from the treated cow.

### 3.1. Theme 1: Expectations of the Farm and Farmer

This theme was the most predominant; this was not surprising given that the respondents were specifically asked if the farmer should treat the cow. While many participants were in favour of the farmer treating the cow, in almost all cases the respondents also voiced a number of caveats, such as “*He should try to keep the treated animal separate from the others, and avoid it from getting near to any streams, ponds, open water etc*.” [Resp. 367]. Other stipulations raised by participants included not allowing the animal’s meat or milk to enter the food chain, observing a lengthy withdrawal period, or treating ‘prudently’ or under the supervision of a veterinarian or other expert. Overall, the majority of participants in this study held the belief that “*antibiotics shouldn’t be used in livestock unless absolutely necessary, as a last resort and only if complete recovery is generally expected*” [Resp. 489].

Many respondents also made a clear distinction between the therapeutic use of antibiotics and their use at sub-therapeutic doses (for growth promotion or prophylaxis), indicating that the former was acceptable and the latter was not. For instance, in the voice of one respondent, “*If the cow is truly sick it should be given antibiotics. However, if antibiotics are only given as a prophylactic, then it is terribly wrong. Not only can they go into the ecosystem to … make resistant bacteria but [the antibiotic] can also be retained in the meat to cause allergic reactions … for someone who is eating it*” [Resp. 331]. Participants also expressed concerns that antibiotics were used all the time. For example, one respondent stated that “*The problems come when farmers give antibiotics all the time, regardless if animals are sick or not. Many farms do this and I think it is a very bad practice. That’s how most antibiotics get into the ecosystem.*” [Resp. 563]. Among those who indicated that the farmer should treat the cow, respondents also frequently commented that to withhold treatment would constitute animal cruelty, and that the farmers “owe” the animal the best-possible care. For instance, “*Refusing to treat the animal is almost cruel*” [Resp. 25] and “*he has a moral obligation to limit the suffering of the cow*” [Resp. 319]. A few participants also felt that a single treatment of antibiotics would not be problematic: “*If I were the farmer, I would use the antibiotics. I don’t think that this one time use would hurt the environment any*.” [Resp. 31].

There were also distinctions made regarding farm size, explicitly between small farms and “*large factory farms giving constant antibiotics*” [Resp. 324]. For example, one participant stated that, “*If a farmer is engaged in factory farming … the farmer should reform his practices rather than utilize antibiotics. However, I am much more sympathetic to farmers who raise their livestock in a humane way and would say that the limited and judicious use of antibiotics would be acceptable in their case*” [Resp. 440]. 

Reactions to the question of what the farmer should do varied, with some respondents answering from the individual animal perspective and others focusing on animal agriculture from a more holistic or abstract perspective. For instance, the following response specifically addressed our question as a one-time event affecting one farmer and one cow. “*If I were the farmer, I would use the antibiotics. I don’t think that this one-time use would hurt the environment any*” [Resp. 31]. Other participants, however, felt that antibiotic use was a systemic challenge facing animal agriculture. For instance, one respondent said “*This scenario isn’t reasonable as it doesn’t pertain to why people are upset that farmers are using antibiotics on farm animals in the USA. They are not using them specifically to fight off infections, but instead feed them a daily mixture because the antibiotics have a side effect of making the animals fatter*” [Resp. 994].

Some participants who were explicit and favoured treatment questioned whether the farmer had the appropriate knowledge of the regulations regarding the environmental impact of antibiotic usage, even questioning whether they had the correct training. For instance, one participant stated “*The farmer should administer the antibiotic to the animal in order to cure it, but he needs to follow the rules of proper dosage and conditions. As a farmer, he’s not equipped to judge the environmental impact by himself, he has to rely on governmental policies as a guidance*” [Resp. 74]. Interestingly, this participant also said “*As long as the antibiotics [sic] is legal, it’s his prerogative to use it.*”.

Many participants also indicated their desire that the farmer take on strategies that focus on prevention and addressing the root cause of the disease. Comments centred on the need for improved management practices, whereby issues such as the overcrowding of animals and the provision of appropriate diets were emphasized. Interestingly, the majority of these comments were intertwined with suggestions of, or questions about, alternatives to antibiotics. As one participant stated “*The farmer needs to learn what conditions lead to animals needing antibiotics … These conditions should be changed and other methods of treating illness should be studied*” [Resp. 60]. The issue of cleanliness was clearly on the minds of many, with specific mention of “*constantly disgusting environments*” [Resp. 324], and that the prophylactic use of antibiotics “*allows them [farmers] to keep these animals in small confined areas, sitting around in there [sic] own waste all day without worry of infection or the need for sanitation*” [Resp. 994].

Despite the numerous caveats and concerns, respondents also alluded to the economic factors involved in the decision of whether the farmer should treat the cow. Some participants indicated that the decision is a ‘right’ that lies exclusively with the farmer. “*Despite the environmental implications this could have, the farmer has every right to give his animal antibiotics. Animals are an investment. Therefore, he should treat his animal to protect his investment*” [Resp. 565]. Some participants also acknowledged that farming was a business with a high degree of uncertainty. “*Farming is a hard life with not a lot of reward, so I doubt that the farmer could afford to lose his cow. This cow is vital to his business*” [Resp. 189]. 

Some participants who received the long text discussed that, if the antibiotics were excreted, was there a possibility of treating the manure or somehow limiting the spread of the antibiotics following application of the manure. For instance, one participant stated “*This part of the farm [where the treated cows are housed] would be retrofitted with special plastic sheets that would be placed under the ground. The role of this is to stop manure waste from leaching into the groundwater. All the manure from the affected animal then should be properly disposed of*” [Resp. 5]. The concept of isolating the treated cow was suggested several times. “*What if the cow was sequestered somewhere such as a ‘quarantine barn’?*” [Resp. 92]. One participant went further, suggesting that “*all manure and manure particles are stopped from landing on the earth’s surface and disposed of in hazardous waste containers*” [Resp. 148].

### 3.2. Theme 2: Consumer-/Citizen-Focused Concerns

Within this theme, many participants commented on the potential impacts of treating cows with antibiotics on human health, including the food products coming from the animal. Participants also emphasized the potential for the development of antibiotic-resistant bacteria. For instance, “… *antibiotics should not be given to farm animals because it gets into both our water supply and the meat we eat. It is unhealthy for humans to have this level of exposure to antibiotics because it can create super bacteria*” [Resp. 186]. Others mentioned the risk to those allergic to antibiotics. One respondent indicated that the farmer should not administer the antibiotic because “*It will remain in the animal’s muscles which are used for food and antibiotics have a wide range of negative effects on people. Many people are allergic to many different kinds of antibiotics, some severely so*” [Resp. 358].

A few respondents referenced a desire for labelling of the resulting food products if the animals had received antibiotics, or transparency in marketing to allow the consumer to choose. Interestingly some respondents indicated a desire for more stringent (or, in the words of one participant, “…dynamically changing”) regulation of the use of antibiotics “*…set by a governing body which takes human antibiotic resistance, the ecosystem, food supply, and everything else that is impacted by the...use of antibiotics in livestock*” [Resp. 424].

### 3.3. Theme 3: The Environment

The respondents receiving the short text rarely made reference to the environment, but some did. Respondents receiving the long text (with or without the video) frequently made mention of the long-term impacts of antibiotic usage and suggested that environmental protection should be also be prioritized. Others conflated the issue of antibiotics in soil with concern about other chemical and pharmaceutical compounds. “*There are way too many chemicals and drugs in our water and soil, and therefore our bodies already, that adding to that amount is a bad idea.*” [Resp. 248] Some participants also stated that the farmer has an inherent duty to protect the environment. “*I think that the farmer has a duty to the environment and to its inhabitants to not use antibiotics known to affect the ecosystem*” [Resp. 942], “*I think the farmer should not give the animal antibiotics. The negatives are too numerous (bad for environment, antibiotic resistance in human population) just so a farmer can make a little bit more money*” [Resp. 488], and, finally, “*I feel it is a moral obligation to not mess with the ecosystem that we have*” [Resp. 681].

### 3.4. Theme 4: Trade-Offs

The concept of trade-offs between the individual animal or farmer and the greater societal good were also raised by some participants, especially among those participants offered the longer text with or without video. Most respondents acknowledged the complexity of the question, referencing more than one theme in their replies. For instance, one individual that said that the farmer should not treat the cow stated “*I have such mixed emotions about this issue. While I feel it is necessary to cure the animal (strictly for humanitarian reasons) I am aware of the dangers of giving animals antibiotics, which then end up in the meat we eat and the water that we drink. Antibiotic resistance in humans is a potentially life-threatening situation*” [Resp. 391]. Another said “*Honestly, that video shed some light on something I hadn’t thought of before. I think the farmer would choose to medicate the cow, because cows are expensive, however, this puts the environment at risk for bacteria that are immune to antibiotics*” [Resp. 413]. Some respondents acknowledged this trade-off but voted the other way. “*Administer the antibiotic. A tiny risk of it ever impacting anyone even slightly versus the definite risk of an animal suffering = the animal shouldn’t be left to suffer*” [Resp. 79].

## 4. Discussion

Participants appeared less supportive of treating the sick cow with antibiotics if they were given more information as to the downstream effects on the environment, antimicrobial resistance, and human health via text, and support decreased further with the addition of video information. That the provision of additional information in the form of images or videos can alter people’s views on particular issues has long been known [40]. For example, in a Canadian study, also based on a convenience sample, on the acceptability of gestation stalls for sows, participants became more supportive of group housing after accessing information, and commented specifically that the visual information provided changed their views [41]. Photographs or videos may be perceived as particularly useful by participants who are removed from the day-to-day practices used in farm-animal production systems [41,42].

### 4.1. Distinction between Therapeutic and Nontherapeutic Uses 

The distinction made by many of our respondents regarding the acceptability of the therapeutic use of antibiotics but concerns regarding sub-therapeutic use is consistent with previous reports. A convenience sample of participants surveyed by Cardoso et al. [34] on the topic of dairy farming rejected the use of antibiotics, hormones, or other chemicals for the purposes of increasing milk production, but many stated that animals should be treated if they are sick. A similar finding, also using a convenience sample, was also reported by Sato et al. [35], who surveyed U.S. citizens on the ideal characteristics of a pig farm. Likewise, a recent systematic review of 80 published studies on citizen and consumer attitudes toward animal welfare makes clear that in those (mostly European) studies, lay people were more concerned with the overuse and misuse of antibiotics, but also generally accepted that antibiotic use cannot be entirely avoided [43]. 

Not surprising given that about 70% of our respondents were 34 years or younger in age, there was a preference for organic products compared with other recent surveys [44]. Likewise, the increased concern regarding the use of antibiotics on dairy farms by those who self-identified as preferring organic food was not unexpected given that human health concerns have been listed as a top priority for this type of consumer [45,46]. Previous research reported that most organic consumers also consider other ethical issues, such as animal welfare, when making decisions (see review by Schleenbecker and Hamm [47]). We encourage more work using representative samples to investigate how dietary preferences affect decision-making regarding the use of antibiotics in dairy farming.

Many respondents clearly perceived that the nontherapeutic use of antibiotics is routine in dairy production systems. This may, in part, be a consequence of the often-used framing used by media linking antibiotic resistance crises to (all of) animal agriculture in media reports (see Steede et al., [48]). In the dairy industry, the administration of antibiotics is primarily therapeutic [49], with the primary prophylactic use in cows during cessation of lactation (‘dry off therapy’), a practice that involves a single injection of antibiotic that every cow experiences on average once per year, to treat subclinical infections and prevent new infections at calving. In North America, dry off therapy is used in 93.0% [50] and 98% [51] of cows in the U.S. and Canada, respectively. Antibiotic uses for dry off therapy are designed to prevent systemic absorption (they are, however, retained in the udder and captured in milk the first days of the following lactation). 

The condemnation of ‘routine feeding’ of antibiotics from the participants in this survey is consistent with other reports of lay or consumer opinions [34,35,52]. In an online survey of primary household shoppers, 66% said they would support a hypothetical proposal to allow antibiotic use only for the treatment of disease [53]. While data on the use of antibiotics in animal agriculture in the U.S. are scarce and imprecise, certainly the use of medically important antibiotics for the promotion of growth or feed efficiency has abated since the adoption of the Veterinary Feed Directive (VFD), which became effective on January 1, 2017. The VFD is an initiative of the U.S. Food and Drug Administration (FDA) intended to promote the responsible use of medically important antibiotics in food-producing animals by phasing in veterinary oversight of the use of antibiotics that were formerly available over-the-counter. The VFD mandates authorization by a veterinarian, in the context of a veterinarian–client–patient relationship, for the use of medically important antibiotics in animal feed or water. Similarly, in the Canadian province of Quebec, antibiotics can only be dispensed to farmers under veterinary prescription or supervision [19].

To dairy and livestock producers, the distinction between growth promotion and prophylactic uses of antibiotics is significant [54,55], but this survey suggests consumers are no more accepting of the latter than they are of the former. The finding that respondents coupled antibiotic use with attempts to increase the growth rate of animals rather than to solely treat certain diseases is not universal to the U.S., as it was also reported in both Italy and Germany [52]. Many participants indicated their desire that the farmer focus on identifying and eliminating the root cause of the disease, with frequent mention of dirty conditions or overcrowding. Our results and those of others [53,56,57,58] imply that the prophylactic use of antimicrobials in livestock will receive little support from consumers and concerned citizens. Over the last decade, the relationship between animal agriculture and society has shifted, resulting in the rejection of some previously common management practices (e.g., tail docking in dairy cattle; [59]). Giving up practices that do not resonate with societal values may undermine the social license provided to animal agriculture. 

Many working within agriculture argue that once consumers are educated regarding the nuances of agricultural practices, support for current practices will follow. However, consumer education does not guarantee acceptance [44,60]. For instance, Ventura et al. [61] showed that despite increasing lay citizens’ knowledge about dairy farming, confidence in the industry was eroded once individuals became aware of certain practices (e.g., cow–calf separation, zero grazing). Similarly, Brazilian citizens with little prior knowledge of dairy farming provided information on the practices of total confinement (zero grazing) and cow–calf separation were more likely to reject these practices once made aware [60]. The fact that certain farm practices fail to resonate with the general public’s values may contribute to increased levels of distrust in farming once the public becomes aware of them [62].

### 4.2. Nuanced Acknowledgement of Trade-Offs

The responses from the open-ended questions in the current survey revealed sympathy for the farmer with recognition that farming is often associated with challenges both economic and associated with farming as a way of life. Benard and de Cock Bunning [63], using a focus group approach, asked Dutch pig farmers and urban citizens to reflect on the perspectives of the other stakeholder. While the farmers staunchly maintained their own perspective and defended their practices, urban citizens recognized pig farming as being hard while expressing the hope that farmers would strive for improved practices [63]. Using a Mechanical-Turk-generated convenience sample, Cardoso et al. (2016) indicated that U.S. citizens, of which the majority were millennials (<35 years of age), stated that profitably was a primary characteristic of an ideal U.S. dairy farm [34]. Similarly, our results, albeit also based on a convenience sample, suggest that the lay public appear to be open to accepting that farming is complex and that it must be economically viable.

Many respondents acknowledged and discussed potential solutions in a nuanced fashion. Somewhat surprising was that regulation of the use of antibiotics was mentioned by few respondents (<5%), with the majority placing more focus on ensuring the farmer sought veterinary advice or had accurate information on the necessary withholding times of the drugs. Moreover, even fewer respondents mentioned labelling food products regarding antibiotic use in their production, despite recent discussion in the media and in the academic literature [64] on the labelling of food products containing genetically modified organisms. 

Respondents who mentioned the size of the farm indicated greater consideration for small farms than for large farms. This is consistent with other surveys [34,43], but the effect of farm size on animal welfare is complex. In a recent review, Robbins et al. [62] found little evidence in support for any simple relationship, negative or positive, between farm size and animal welfare. In fact, the available evidence indicates that larger farms likely permit more specialized and professional management of animal health [62], which may facilitate improved antibiotic practices. A romantic view of agriculture that includes small family farms [see Fraser [65]] may reflect a desire to resurrect farming practices of the past, when antibiotics were not available and farming was viewed as a way of life rather than a business [66]. It should also be noted that, as manure is commonly used as fertilizer and spread widely on fields to produce crops, its application will have a widespread impact on soil and water health (reviewed by Rayne and Aula [67]).

Despite the largely urban and suburban residence of our convenience sample of respondents, some asked about or suggested technological interventions that could prevent environmental contamination arising from the use of the antibiotics, such as manure treatment. Although the segregation of manure from treated farm animals is not common practice on most farms, there is a growing body of evidence suggesting that composting and anaerobic digestion can degrade many antibiotics [68,69,70]. Ray et al. [71] reported that nearly the entire dose of a mastitis antibiotic treatment is excreted within 5 days after administration. We see the use of manure treatment methods that are effective in degrading antibiotics becoming increasingly viable, particularly as the total amount of manure requiring treatment is small relative to the total manure produced on the farm. Although the capture of manure in this manner is not practical at this time, we strongly encourage future research to identify practical methods that allow for the capture of manure from treated animals.

The lack of many respondents specifically making reference to the environment was similar to the results of Cardoso et al. [34]. However, in both the present study and in the latter study, those that did comment on the environment clearly indicated their desire that agriculture employ practices that reflect a high level of responsibility towards the environment. These competing priorities between the farmer’s obligations to treat the cow verses the potential downstream effects on the environment are the crux of the issue; namely, how does society balance human health and ecosystem demand with the animal welfare challenge of untreated disease? In European countries, these competing concerns are addressed by the implementation of increasingly strict limitations on the use of antibiotics [18]. In the U.S., the diagnosis of disease and initiation of treatment is typically conducted by farmers, particularly on small farms [72], and the threshold of the perceived severity of disease that requires antibiotic treatment varies amongst them [73]. In Sweden and most other European countries, however, antibiotics may only be administered by a veterinarian and the presence and identity of the pathogen must be confirmed [74].

## 5. Limitations

The convenience sample used in this study should not be considered representative of any one state or, indeed, the entire U.S. population. The topic of antimicrobial resistance is also a complex topic, and so it is likely that for most of our participants, who had little knowledge of animal agriculture, their responses were based on what they believe they know rather than from a position of scientific fact. The findings herein also represent the ideas of a group of primarily urban and suburban U.S. citizens of a variety of ages, both sexes, and with a relatively high education level. Our recruitment strategy of including antibiotic usage in the short running title where participants could select which survey(s) they wanted to complete may have biased our sample towards individuals interested in this topic. We also recognize that using the AmazonPrime website no doubt biased our findings, as individuals were required to hold an account and be computer-literate. We recognize that our participant sample was also biased towards individuals under the age of 35–40, but, as suggested above, this could also be seen as a benefit given that they include the generations that will likely play an increasing role as their buying power increases.

## 6. Conclusions

Among those expressing an opinion of whether to treat a sick cow, many respondents urged the farmer to treat the cow. When presented with additional information (both written and video) specifically describing the link between antibiotic use and the potential spread of antibiotic resistance, support for antibiotic therapy appeared to wane. Respondents were overall critical of sub-therapeutic or prophylactic antibiotic use and showed a strong desire that agriculture employ practices that limit the outflow of antibiotics from the farm into the broader environment. Efforts focusing on the prevention of disease without reliance on nontherapeutic doses of antibiotics must continue. The nuanced commentary by some participants indicated that despite the overarching desire that the farmer treats the animal, they also believed that efforts should focus on segregating the urine and manure. The design of sick pens that allow for the capture of treated manure seems a promising avenue of research. Engagement with the public on contentious issues such as antibiotic use supports the development of practices that resonate with all stakeholders and is critical for the continuation of the social license to operate [75]. 

## Figures and Tables

**Table 1 animals-13-02913-t001:** Demographics of U.S. participants in an online survey focused on attitudes about therapeutic use of antibiotics in animal agriculture (*n* = 779).

Demographic Category	Variable	%
Age	18–24	21.8
	25–34	45.4
	35–44	15.0
	45–54	11.2
	55 and above	6.6
Gender	Identified as female	46.3
Level of education	High school	25.2
	Associate’s degree	15.5
	Bachelor’s degree	41.5
	Graduate/Professional	14.3
	Other	3.6
Political affiliation	Democrat	43.5
	Republican	12.8
	Independent	28.2
	Libertarian	6.0
	Other or no reply	9.4
Dairy product consumption preference	Organic	44.9
	Conventional	45.3
	Don’t consume dairy	9.8
Area of residence	Rural	20.7
	Urban	23.3
	Suburban	54.0

## Data Availability

The following supporting information can be downloaded at: Dataverse https://borealisdata.ca/dataset.xhtml?persistentId=doi:10.5683/SP3/1HCPGA.

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
