# Peer review of "To Treat or Not to Treat: Public Attitudes on the Therapeutic Use of Antibiotics in the Dairy Industry—A Qualitative Study"

_animals, 2023, doi:10.3390/ani13182913_

Round 1
Reviewer 1 Report
31-36: The importance of antibiotic resistance is the non-judicious use of antimicrobials. This is important to address
59-63: Mastitis is a poor example for antibiotic use. Many pathogens that cause mastitis do not respond antibiotics and other therapies are recommended
67: Rather than production, growth would be a better term to specifically describe the purpose
86-96: How were participants initially contacted and/or observe that individuals were being recruited for survey completion? It sounds like the population may have already been aware that Amazon supports surveys and a small renumeration is provided. If it is not already, it should be noted that the participant recruitment may be a limitation because it selected individuals that were potentially more “vocal” or apt to share their opinions than the general population.
103-119: The information provided to the participants was vague and provided limited information. Can the authors please address if this was intentional? And if so, the purpose for providing such little information to particpants
105-107/117-119: Please provide the animated video or its script and visuals be provided as supplementary material
130-132: Were any other explanatory variables included in the model?
132-138: Why was a Chi-Square Analysis performed instead of a more sophisticated model?
139-146: Please define the data analysis method (e.g., content, thematic, discourse, etc.)
139-166: Please include a positionality statement for the authors and qualitive response reviewers
282-327: The selected quotations seem very one sided and support a single ideology within the theme. Were there opposing viewpoints? It seems these should also be referred to, as we can expect not all participants had the same feelings toward the subject
329-334: This interpretation is technically accurate. However, it feels misleading. It should be addressed that a larger percentage of respondents in the short and long text treatments said to treat vs. not treat the cow, and long text-with video had an equal response to treating vs. not treating the cow. Additionally, within the discussion it will be important to address that between a third and half of respondents said they were unclear or unsure.
401-402: Miswritten?
423: Please provide age range for millennials
445-456: It may be important to address that manure is commonly used as fertilizer and spread widely on fields to produce crops. Manure that originated at a single farm will have a widespread impact on soil and water health
Author Response
Reviewer 1
We thank the reviewer for their time and insightful comments. We have incorporated them and believe that the manuscript is much improved. Below we provide a detailed response to each comment.
31-36: The importance of antibiotic resistance is the non-judicious use of antimicrobials. This is important to address
AU: We have inserted a sentence on Line 47-49 stating that the role of non-judicious use of antimicrobials to antibiotic resistance. “It is well recognized that non-judicious use of antibiotics are the greatest contributor to antimicrobial resistance (reviewed by Zaman et al., 2017). That said,…”
59-63: Mastitis is a poor example for antibiotic use. Many pathogens that cause mastitis do not respond antibiotics and other therapies are recommended
AU: We recognize that it this may be the case but our intent was simply to give an example of where a cow may be given antibiotics to combat illness.
67: Rather than production, growth would be a better term to specifically describe the purpose
AU: we have changed this to say ‘production and growth’
86-96: How were participants initially contacted and/or observe that individuals were being recruited for survey completion? It sounds like the population may have already been aware that Amazon supports surveys and a small renumeration is provided. If it is not already, it should be noted that the participant recruitment may be a limitation because it selected individuals that were potentially more “vocal” or apt to share their opinions than the general population.
AU: We have now included this in the limitations section the following given our use of the AmazonPrime site as the provider of the survey Line 469-470 …” We also recognize that using the AmazonPrime website no doubt biased our findings as individuals were required to hold an account and be computer literature.” However, we do also provide references that support the use of these types of providers in the methods on line 102-106: “Participants were invited online via Mechanical Turk (MTurk, www.mturk.com) a method of data collection that is relatively quick, easy and yields a diverse (Buhrmester et al., 2011) and attentive (Hauser and Schwarz, 2016) pool of subjects, compared to more traditional subject pools, and has been reported to provide high-quality and reliable data (Saunders et al., 2013).”
Throughout the paper we also continually remind the reader that our samples is based on a convenience sample.
103-119: The information provided to the participants was vague and provided limited information. Can the authors please address if this was intentional? And if so, the purpose for providing such little information to participants.
Au: Our intent was to provide enough information written in such a way that it would be accessible to a broad cross section of society that is not familiar with the dairy industry. Our objective was to elicit their top-of-mind response when presented this information. We also did not seek a representative sample and thus now explicitly state that our results are not generalizable across all publics. We also recognize that our survey did not allow the participants to ask for more information which can also be viewed as a potential limitation. We now explicitly state this in the study on line 460-464: “The convenience sample used in this study should not be considered representative of any one state or indeed the entire U.S. population. The topic of antimicrobial resistance is also a complex topic and so it is likely that for most of our participants who had little knowledge of animal agriculture their responses were based on what they believe they know rather than from a position of scientific fact.”
105-107/117-119: Please provide the animated video or its script and visuals be provided as supplementary material
AU: Done. We have provided the video in the supplementary material as requested.
130-132: Were any other explanatory variables included in the model?
AU: Upon reflection of the comments made by both reviewers we have elected to remove the quantitative results from this study. This was done as only 61% responded directly to the question of whether the farmer should or should not treat the cow with antibiotics by answering with a definitive yes/no. We have also changed the title to clearly state that this is a qualitative study. We now explicitly state this on Line 138-147.
132-138: Why was a Chi-Square Analysis performed instead of a more sophisticated model?
AU; As stated above based on the comments by the various reviewers we have removed the quantitative results as only 61% responded directly to the question of whether the farmer should or should not treat the cow with antibiotics by answering with a definitive yes/no. We have also changed the title to clearly state that this is a qualitative study.
139-146: Please define the data analysis method (e.g., content, thematic, discourse, etc.)
AU: We have added clarity on this and now state that the qualitative responses were subjected to inductive thematic analyses on Line 151 now state: “data reduction was achieved using inductive thematic analyses (coding of information around the common ideas conveyed).”
139-166: Please include a positionality statement for the authors and qualitive response reviewers
AU: We now provide this on Lines 87-96: “KK grew up on a dairy farm in Connecticut and earned her B.S. in Animal Science at Cornell University, and an MSC and PhD degree at Michigan State and the University of Maryland, respectively. She is a professor in the Department of Dairy Science at Virginia Tech where both her teaching and research focused on environmental issues affecting the dairy industry. MvK grew up on a beef cattle ranch and earned her MSc and Ph.D. in Animal Science from the University of Alberta and UBC, respectively, before working for the feed industry for 7 years before returning to UBC as a faculty member. Her research focus is animal welfare and she co-leads the UBC Animal Welfare Program. MvK has published extensively in the social sciences literature on issues relating to farm animal welfare.”
282-327: The selected quotations seem very one sided and support a single ideology within the theme. Were there opposing viewpoints? It seems these should also be referred to, as we can expect not all participants had the same feelings toward the subject
AU: We have included an additional quote in this section to provide some balance but this arguably opposing viewpoint was rare. See line 214-215 “A few participants also felt that a single treatment of antibiotics would not be problematic: “If I were the farmer, I would use the antibiotics. I don't think that this one time use would hurt the environment any. [Resp. 31].
329-334: This interpretation is technically accurate. However, it feels misleading. It should be addressed that a larger percentage of respondents in the short and long text treatments said to treat vs. not treat the cow, and long text-with video had an equal response to treating vs. not treating the cow. Additionally, within the discussion it will be important to address that between a third and half of respondents said they were unclear or unsure.
AU: Given the feedback from the reviewers and the fact that almost 40% of the respondents did not clearly state whether the animal should be treated or not we have made the decision to remove the quantitative analyses we have also removed this section of the discussion.
401-402: Miswritten?
AU: We are not sure what the reviewer is referring to here.
423: Please provide age range for millennials
AU: added on line 403 “millennials (<35 years of age),..”
445-456: It may be important to address that manure is commonly used as fertilizer and spread widely on fields to produce crops. Manure that originated at a single farm will have a widespread impact on soil and water health
AU: We have added this information on line and added a reference supporting this statement: Line 442-445 “It should also be noted that on manure is commonly used as fertilizer and spread widely on fields to produce crops manure which will have a widespread impact on soil and water health (reviewed by Rayne and Aula 2020). “
Rayne, N.; Aula, L. 2020. Livestock Manure and the Impacts on Soil Health: A Review. Soil Syst. 4, 64. https://doi.org/10.3390/soilsystems4040064
Reviewer 2 Report
The subject is of utmost importance from the One Health perspective concept and from the view-point of its extreme complexity.
Nevertheless, the authors approach is very narrow, and seems directed from the very beginning of the study. The authors admitted the study might be biased due to a selected category (educated, urban-suburban) of responders, rather than random.
The study pattern, by providing the study subjects with limited unidirectional information on the investigated event (Imagine that a farmer has an animal that he/she is raising for food and that the animal gets sick. The farmer knows that administering antibiotics known to treat this illness could improve the animal's condition. or the long version, adding the information on antimicrobial resistance plus a video), involved a subjective approach, further biased the results.
Which would have been the answers of the responders, knowing that the problem is far beyond treating or not treating the cow, thus protecting the environment and human health from antibiotic residues and that the source for antimicrobial resistance could be as well the humans as the environment?!
See the literature mentioned in the attached list.
It is quite logical, on behalf of the responders, to agree with use of antibiotics to save the cow, for various reasons (humane, economic or others) and disagree with growth promoter use of those - the legal measure to give up such purpose has been long in use in numerous parts of the world. Nevertheless, from consumer's point of view, the opinions of the responders are highly variable, concerning less the safety of the consumed dairy products and more the methodology to prevent antibiotic spread in the environment, disregarding the lack of their competence to indicate such measures.
Finally, in the conclusions, the authors pick up some suggestions of the non-professional responders (i.e., "The design of sick pens that allow for capture of treated manure seems a promising avenue of research." - such sick pens exist as isolation units, for a long time; "efforts should focus on segregating the urine and manure" - hardly possible on farms, where sewage is jointly collected and then filtered and composted).
"Engagement with the public on contentious issues such as antibiotic use supports the development of practices that resonate with all stakeholders and is critical for continuation of the social license provided to animal agriculture by society." - sounds presumptuous, without a sound scientific background.

Author Response
Reviewer 2
We thank the reviewer for their time and insightful comments. We have incorporated them and believe that the manuscript is much improved. Below we provide a detailed response to each comment.
The subject is of utmost importance from the One Health perspective concept and from the view-point of its extreme complexity.
AU: Thank you!
Nevertheless, the authors approach is very narrow, and seems directed from the very beginning of the study. The authors admitted the study might be biased due to a selected category (educated, urban-suburban) of responders, rather than random.
AU: Based on the reviewer comments and the fact that our sample was not representative we have elected to remove the quantitative results of this study and focus on the qualitative responses. Qualitative work is not meant to be generalizable but rather provide in depth insights on the views and beliefs of the individuals who participated in the study. We have now expanded the discussion to include a limitations section where we caution the readers on this issue. Lines 461 - 477: “The convenience sample used in this study should not be considered representative of any one state or indeed the entire U.S. population. The topic of antimicrobial resistance is also a complex topic and so it is likely that for most of our participants who had little knowledge of animal agriculture their responses were based on what they believe they know rather than from a position of scientific fact. The findings herein also represent the ideas of a group of primarily urban and suburban US citizens of a variety of ages, both sexes and with a relatively high education level. Our recruitment strategy of including antibiotic usage in the short running title where participants could select which surveys they wanted to complete, may have biased our sample towards individuals interested in this topic. We also recognize that using the AmazonPrime website no doubt biased our findings as individuals were required to hold an account and be computer literature. We recognize that our participant sample was also biased towards individuals under the age of 35-40 but as suggested above could also be seen as a benefit given that they include the generations that will likely play an increasing role as their buying power increases.”
The study pattern, by providing the study subjects with limited unidirectional information on the investigated event (Imagine that a farmer has an animal that he/she is raising for food and that the animal gets sick. The farmer knows that administering antibiotics known to treat this illness could improve the animal's condition. or the long version, adding the information on antimicrobial resistance plus a video), involved a subjective approach, further biased the results.
AU: As stated above we have removed the quantitative results and only provide the results regarding the qualitative responses. We clearly state that our results are based on a convenience sample and are not generalizable across all publics.
Which would have been the answers of the responders, knowing that the problem is far beyond treating or not treating the cow, thus protecting the environment and human health from antibiotic residues and that the source for antimicrobial resistance could be as well the humans as the environment?!
See the literature mentioned in the attached list.
AU: Thank you for the citations. We have used some of them in the introduction.
It is quite logical, on behalf of the responders, to agree with use of antibiotics to save the cow, for various reasons (humane, economic or others) and disagree with growth promoter use of those - the legal measure to give up such purpose has been long in use in numerous parts of the world. Nevertheless, from consumer's point of view, the opinions of the responders are highly variable, concerning less the safety of the consumed dairy products and more the methodology to prevent antibiotic spread in the environment, disregarding the lack of their competence to indicate such measures.
Finally, in the conclusions, the authors pick up some suggestions of the non-professional responders (i.e., "The design of sick pens that allow for capture of treated manure seems a promising avenue of research." - such sick pens exist as isolation units, for a long time; "efforts should focus on segregating the urine and manure" - hardly possible on farms, where sewage is jointly collected and then filtered and composted).
AU: We recognize the impracticality of this statement but would like to retain this statement given that agriculture practices are continually evolving. We have however added an additional statement stating that under current practices this may not be possible. Line 438 “Although capture of manure in this manner is not practical at this time, we strongly encourage future research to identify practical methods that allow for the capture of manure from treated animals.”
"Engagement with the public on contentious issues such as antibiotic use supports the development of practices that resonate with all stakeholders and is critical for continuation of the social license provided to animal agriculture by society." - sounds presumptuous, without a sound scientific background.
AU: We have provided a reference that supports this statement. Line 478: “Engagement with the public on contentious issues such as antibiotic use supports the development of practices that resonate with all stakeholders and is critical for continuation of the social license to operate (Moffat et al., 2016). “
Some of the articles tackling the complex issue of antibiotic resistance onset and spread:
Wegener HC. ANTIBIOTIC RESISTANCE—LINKING HUMAN AND ANIMAL HEALTH. In: Institute of Medicine (US). Improving Food Safety Through a One Health Approach: Workshop Summary. Washington (DC): National Academies Press (US); 2012. A15. Available from: https://www.ncbi.nlm.nih.gov/books/NBK114485/
Tao, S., Chen, H., Li, N., Wang, T., & Liang, W. (2022). The Spread of Antibiotic Resistance Genes In Vivo Model. The Canadian journal of infectious diseases & medical microbiology = Journal canadien des maladies infectieuses et de la microbiologie medicale, 2022, 3348695. https://doi.org/10.1155/2022/3348695
Nair, R. R., & Andersson, D. I. (2023). Interspecies interaction reduces selection for antibiotic resistance in Escherichia coli. Communications biology, 6(1), 331. https://doi.org/10.1038/s42003-023-04716-2
Rousham EK, Unicomb L,Islam MA. 2018 Human, animal and environmental contributors to antibioticresistance in low-resource settings: integratingbehavioural, epidemiological and One Healthapproaches.Proc. R. Soc. B285: 20180332.http://dx.doi.org/10.1098/rspb.2018.0332
Iramiot, J.S., Kajumbula, H., Bazira, J. et al. Antimicrobial resistance at the human–animal interface in the Pastoralist Communities of Kasese District, South Western Uganda. Sci Rep 10, 14737 (2020). https://doi.org/10.1038/s41598-020-70517-w
and so many more.
AU; Thank you for these additional references. We have now included most of them in the manuscript.
Round 2
Reviewer 1 Report
Thank you very much for your edits to the paper and for providing the video. One small thing to check, there was no sound with the video on YouTube.